# Social Support among Older Persons and Its Association with Smoking: Findings from the National Health and Morbidity Survey 2018

**DOI:** 10.3390/healthcare11162249

**Published:** 2023-08-10

**Authors:** Ambigga Krishnapillai, Chee Cheong Kee, Suthahar Ariaratnam, Aida Jaffar, Mohd Azahadi Omar, Ridwan B. Sanaudi, Rajini Sooryanarayana, Ho Bee Kiau, Sazlina Shariff Ghazali, Noorlaili Mohd Tohit, Sheleaswani Inche Zainal Abidin

**Affiliations:** 1Department of Family Medicine, Faculty of Medicine and Health, National Defense, University of Malaysia, Sg. Besi 57000, Malaysia; aida@upnm.edu.my; 2Sector for Biostatistics and Data Repository, National Institutes of Health, Ministry of Health Malaysia, Shah Alam 40170, Malaysia; drazahadi@moh.gov.my (M.A.O.); ridwan.s@moh.gov.my (R.B.S.); 3Department of Psychiatry, Faculty of Medicine, Universiti Teknologi MARA (UiTM), Sg. Buloh 47000, Malaysia; suthaharariaratnam@yahoo.com.au; 4Family Health Development Division, Ministry of Health Malaysia, Putrajaya 62590, Malaysia; drrajini@moh.gov.my (R.S.); dr.sheleaswani@moh.gov.my (S.I.Z.A.); 5Klinik Kesihatan Bandar Botanik, Ministry of Health Malaysia, Klang 42000, Malaysia; bkho@hotmail.com; 6Department of Family Medicine, Faculty of Medicine and Health Sciences, Universiti Putra Malaysia, Serdang 43400, Malaysia; sazlina@upm.edu.my; 7Laboratory of Medical Gerontology, Malaysian Research Institute on Ageing (MyAgeing), Universiti Putra Malaysia, Serdang 43400, Malaysia; 8Department of Family Medicine, Faculty of Medicine, Universiti Kebangsaan Malaysia, Kuala Lumpur 56000, Malaysia; laili@ppukm.ukm.edu.my

**Keywords:** social support, older persons, smokers, NHMS 2018, community survey

## Abstract

Background: Globally, the average age of the world’s population of older people continues to rise and having a good social support network becomes increasingly relevant with the aging populace. Overall, in Malaysia, social support prevalence was low among older persons. This study was conducted to determine the association between social support and smoking status among the older Malaysian population. Methods: Data were obtained from the National Health and Morbidity (NHMS) 2018 survey on the health of older Malaysian adults and analyzed. This cross-sectional population-based study used a two-stage stratified random sampling design. Sociodemographic characteristics, smoking status, and social support data were collected from respondents aged 60 years and more. A validated Malay language interviewer-administered questionnaire of 11-items, the Duke Social Support Index, was utilized to assess the social support status. A multivariable logistic regression analysis was used to assess the association of social support and smoking status among the respondents. Results: The prevalence of good social support was significantly higher among the 60–69 years old (73.1%) compared to the ≥80 years old respondents (50%). Multivariate logistic regression analysis showed that respondents aged ≥80 years old were 1.7 times more likely to have poor social support compared to those aged 60–69 years. Respondents with no formal education were 1.93 times more likely to have poor social support compared to respondents who had tertiary education. Respondents with an income of <MYR 1000 were 1.94 times more likely to have poor social support compared to respondents with an income of >MYR 3000. Former smokers had good social support compared to current smokers (73.6% vs. 78.7%). For current smokers, they had poor social support, which is almost 1.42 times higher than that for non-smokers. Conclusion: There was poor social support among older people who were current smokers, had an increased age, had no formal education and had a low income. The findings obtained from this study could assist policymakers to develop relevant strategies at the national level to enhance the social support status among older smokers and aid in their smoking cessation efforts.

## 1. Introduction

The average age of the world’s population of older people continues to rise swiftly. The global population aged 60 years or over in 1980 was 382 million and this doubled to 962 million people in 2017 [1]. The number of older people is expected to double again by 2050; it is projected to reach nearly 2.1 billion [1]. In 2050, projections reveal that there will be more older persons aged 60 or over than adolescents and youth of ages 10–24 (2.1 billion versus 2 billion) [1]. Persons aged 80 years or over are projected to increase globally by more than threefold between 2017 and 2050, rising from 137 million to 425 million [1]. Two thirds of older people live in developing regions, wherein their numbers are rising faster than in the developed regions [1]. Similarly, a rapid growth spurt is occurring in the middle-income countries like Malaysia. At present, Malaysia has a population of 32.7 million and 7% of this population consists of older persons aged above 65 years [2]. The older population in Malaysia increases annually, making it an ageing nation [2]. As the population ages, having good social support becomes increasingly relevant to support this vulnerable group.

While previous research has explored various determinants of smoking in older adults, the role of social support in this context remains underexplored. Social support is defined as an exchange of resources between two individuals, the provider and the recipient, with an aim to enhance the well-being of the recipient [3]. It moderates the effects of health-related strain on mental health in older persons. In addition, it shields the ill effects of stress on both mental and physical health. Good social support is also associated with reduced mortality, reduced depression, and improved well-being in older people [4,5,6,7,8]. Therefore, it is important to establish social networks for the elderly. Increasing the social support and networking among the elderly provides avenues for them to actively participate and engage with their communities. The concept and theories of social support are not within the scope of this paper. Social support encompasses emotional, instrumental, and informational assistance provided by social networks, including family, friends, and community [9]. It plays a vital role in promoting overall well-being, especially among older individuals [10]. Numerous studies have shown that high levels of social support are associated with improved mental and physical health outcomes. However, the impact of social support on smoking behavior among older adults remains a topic of interest.

Social support prevalence was found to be reduced among older persons, which was 30.76% [11]. However, among Malaysian adolescents the prevalence of social support was higher, 55.4% [12]. In 2020, Mahmud et al. [13] found that having low income, being single, not having depression, absence of activities of daily living and having dependency in instrumental activities of daily living were important factors related to perceived social support among Malaysian older adults. Several studies have reported a negative association between social support and smoking prevalence among older persons, such as Kendzor et al. [11], who found that higher levels of social support were associated with lower smoking prevalence among older adults. Similarly, a study by Holahan et al. [12] revealed that older adults with greater social support were less likely to smoke. Different types of social support have been examined in relation to smoking cessation among older persons. Emotional support, instrumental support, and informational support have shown varying degrees of influence on smoking cessation [13]. 

Despite extensive research on the association between social support and health outcomes in various countries, there is a noticeable research gap when it comes to understanding the specific relationship between social support and smoking behavior among older adults in Malaysia. 

This research gap is particularly important to address, given the persistent public health concern of smoking, especially among the older population. Thus, there is a need to delve deeper into the association between social support and smoking behavior in the context of older adults. By examining factors such as age, education, income, and smoking status, we can gain a comprehensive understanding of how these variables shape social support outcomes for this population. Such insights are crucial for the development of targeted interventions and support systems that promote healthy aging and reduce smoking prevalence among older individuals. 

The composition of social networks has been found to impact smoking behavior among older persons. A study conducted on older adults with larger social networks of non-smokers showed that they were more likely to quit smoking [14]. Conversely, being surrounded by smokers in the social network was associated with increased smoking prevalence among older individuals. Some studies have examined gender differences in the association between social support and smoking among older persons. Positive social support discourages smoking in middle-aged and older postmenopausal women [15]. Thus, social influences are important correlates of smoking status among older women.

Social support is also an extremely vital determinant in the tobacco cessation planning among current smokers. Risk factors such as alcohol consumption and smoking pose additional health challenges and increase the vulnerability of the elderly to various diseases and conditions. Supportive relationships provide emotional comfort, stress reduction, and a sense of belonging, which may reduce the need for smoking as a coping mechanism. Cigarette smoke contains many perilous compounds that can cause morbidity [16]. In Malaysia, the prevalence of smoking in the older population was under-reported, in contrast to smoking studies among adolescents [17]. The prevalence rate for current older smokers in Malaysia in 2015 was 11.9% [18]. Tobacco usage is recognized as one of the primary reasons for the rising cardiovascular diseases and untimely deaths in developing countries worldwide [19]. An estimated 71% of lung cancers, 42% of chronic respiratory diseases and nearly 10% of cardiovascular diseases were caused by tobacco smoking [20]. Moreover, the risk of communicable diseases such as tuberculosis and lower tract respiratory infections and decreasing life expectancy [19,20] were also attributed to smoking. Smoking was also associated with divorce, depression, and anxiety [21]. Additionally, good social support is essential when relevant authorities undertake efforts to ban smoking. The aim of this study was to ascertain social support and its association with smoking status among the older Malaysian population. The findings will shed light on the specific dynamics of social support among older adults and provide insights into the factors that influence their smoking behavior. 

## 2. Methods

### 2.1. Data Source and Study Population

Data from the National Health and Morbidity (NHMS) 2018 survey on the health of older Malaysian adults who were residing in the community and were non-institutionalized were analyzed. It provided valuable insights into the health and well-being of Malaysia’s elderly population. The NHMS 2018 was a cross-sectional study, involving the Malaysian population in a nationally representative survey conducted by the Institute for Public Health, National Institutes of Health, Ministry of Health, Malaysia. This survey was funded by the Ministry of Health, Malaysia, in which data collection was conducted between August 2018 and October 2018. Malaysia recognises persons aged 60 and above as elderly [9]. The target population of this survey were the pre-elderly aged between 50 years and 59 years as well as older people aged 60 years and above. There were 3959 older people recruited into the survey. We included 3923 older people aged 60 years and above in the final analysis after excluding non-citizens and missing data due to incomplete data entry (1%). The survey also indicated that a considerable number of older adults engaged in health risk behaviors such as smoking and alcohol consumption. These risk factors pose additional health challenges and increase the vulnerability of the elderly to various diseases and conditions. The findings from the survey sheds light on the challenges and opportunities in addressing the healthcare needs of Malaysia’s aging population in terms of social support. The Medical Research and Ethics Committee of the Ministry of Health Malaysia had approved this study (NMRR-17-2655-39047).

### 2.2. Sampling Method

The Department of Statistics of Malaysia provided the sampling frame which was updated in 2017. Malaysia is geographically divided into about 83,000 enumeration blocks (EBs). In each EBs, there were approximately 80 to 120 living quarters (LQs) with an average household size of 4. This survey applied multiple-stage cluster sampling. First, Malaysia was stratified into states (primary stratum). Second, each state was further stratified into urban and rural areas (secondary stratum). The total number of EBs selected were proportionate to the population size. A total of 60 EBs from urban areas and 50 EBs from rural areas were randomly selected, which consisted of 5636 LQs. All the household members aged 50 years and above were recruited into the study. The detailed methodology for this survey was reported elsewhere [11].

### 2.3. Data Extraction

The data extracted from the NHMS 2018 were respondents’ sociodemographic characteristics, smoking status, and social support. The sociodemographic characteristics collected were a respondent’s current residential area, sex, age, ethnicity, marital status, employment status, monthly income, and education level. We classified smoking status as current smokers, former smoker and non-smokers. Additionally, the respondent’s residential areas were divided into 2 categories: urban and rural areas. The ethnic groups comprised Malays, Chinese, Indians, Other Bumiputra (indigenous groups in Sabah and Sarawak), and others. Age was categorized into 3 groups: 60–69 years, 70–79 years, and 80 years and above. Respondent’s marital status was categorized into two categories: married and single (single/widow/widower/divorcee). Education was divided into 4 levels: no formal education, primary education, secondary education and tertiary education. The employment status was categorized as either employed or unemployed. Individual monthly income was classified into less than MYR 1000, MYR 1000–1999, MYR 2000–2999 and more than MYR 3000. The smoking status was classified as follows: current smokers are those who currently used any smoked tobacco products daily or less than daily, former smoker are persons who used tobacco products in the past and non-smokers are those who never smoked in the past.

The Duke Social Support Index (M-DSSI), a validated Malay language interviewer-administered questionnaire of 11 items, was used for assessing social support among the elderly [22]. The M-DSSI was chosen for this study because it was a short-scale tool used for the elderly and was supported as an instrument in health promotion strategies, as well as aged care research [13]. The scale was developed by Koenig et al. [23] and subsequently translated into Malay with a Cronbach’s alpha of 0.79 by Ismail et al. [24]. A higher M-DSSI score denotes a high level of social support. The 11-item DSSI consisted of 4 questions which measure the size and structure of social network (social interaction) and 7 questions which assess the respondent’s perceived satisfaction with the behavioral or emotional support obtained from their social network (subjective support). A respondent was identified as having poor social support if his/her score was less than 27 [25]. 

### 2.4. Statistical Analysis

A weighting factor was applied to each individual data to adjust for non-response and probability of selection to account for the complex samples design to ensure sufficient representative of the elderly population in Malaysia. The sociodemographic information (residential area, gender, age, ethnicity, marital status, education level, employment status and monthly individual income received), social support, smoking status (non-smoker, former smoker, and current smoker) were presented in frequencies and percentages. Univariable logistic regression was conducted to determine the associations between sociodemographic characteristics, smoking status, and risk of having poor social support. The variables with significant level less than 0.25 were entered into multiple variable logistic regression analysis. Complex samples logistic regression analysis was applied to determine the association between smoking status and risk of having poor social support among the elderly, while adjusting for other confounders (age, gender, ethnicity, marital status, urban/rural, education level, employment status, smoking habit, income and living status) in the logistic regression model. The model fit was assessed based on the receiver operating characteristics curve and classification of table. There were no significant 2-way interactions or multicollinearity found between the variables. All statistical analyses were carried out at 95% significance level using IBM SPSS statistics for the Window version 26.

## 3. Results

A total of 3923 elderly respondents were included in the study. According to the findings, the elderly population constituted a significant proportion of Malaysia’s total population, reflecting the country’s demographic transition. The survey indicated that most elderly Malaysians were of Malay ethnicity (58.1%), followed by Chinese and Indian populations. Majority of the respondents were in the 60–69 years age group (66.5%), females (51.2%), married (67.8%), from urban areas (73.2%), had a primary education (43.6%), unemployed (75.8%), had an income of less than MYR 1000 (58.2%), non-smokers (74.3%) and not living alone (93.7%) (Table 1).

Prevalence of good social support among the Malaysian elderly was significantly higher among the 60–69 years old compared to the ≥80 years old respondents (73.1%, 95% CI: 69.3–76.5% vs. 50.1%, 95% CI: 41.7–58.6%). Multivariate logistic regression analysis revealed that the respondents aged ≥80 years old had 1.7 times poorer social support (aOR: 1.72%, 95% CI: 1.19–2.48), compared to the respondents aged 60–69 years, after controlling the other predictors in the model. Respondents with no formal education were 1.93 times more likely to have poorer social support compared to respondents with a tertiary education (aOR: 1.93%, 95% CI: 1.13, 3.30). Respondents with an income of <MYR 1000 were 1.94 times more likely to have poorer social support compared to respondents with an income of >MYR 3000 (aOR: 1.94, 95% CI: 1.21–3.13). Former smokers had higher good social support compared to current smokers (73.6%, 95% CI: 67.7–78.7 vs. 65.1%, 95% CI: 58.4–71.2). Current smokers had 1.42 times higher poor social support compared to non-smokers (aOR: 1.42, 95% CI: 1.05–1.91), after keeping the other predictors constant. (Table 2) 

## 4. Discussion

To the best of our knowledge, this is the first paper exploring the role of social support and smoking status in a nationally representative sample of older Malaysians. The findings of this study provide valuable insights into the social support networks among elderly individuals in Malaysia, highlighting the impact of the various socio-demographic factors and smoking. This study revealed that majority of the respondents fell within the 60–69 years age group, which is consistent with the aging population trend observed in Malaysia. 

The prevalence of good social support was 73% in the 60–69 years old compared to 50% in the 80 years and more population. Additionally, the 80 years and more respondents were 1.7 times more likely to be associated with poor social support compared to their younger peers. This highlights an important distinction in social support availability across different age groups of the elderly population. One of the probable reasons for poor social support in the 80 years and above was a lack of spousal support due to illness or death of spouse [26]. In addition, this finding is in line with several studies conducted globally, which have consistently reported a decline in social support networks and resources as individuals age [27]. 

Educational profiling found that respondents with no formal education were twice as likely to have poor social support. This finding is consistent with previous research that has established a positive association between higher education levels and stronger social support networks among older adults [28]. Respondents with an income of less than MYR 1000 were also twice as likely to have poor social support. This latter finding concurs with the literature in which older respondents with lower income were perceived to have poor social support [13]. In addition, poor social networks and low social support were more frequent in people of the lower socioeconomic level [29]. Due to this reason, a higher need for social support is warranted among those with lesser income because of the increased financial pressure.

The prevalence for non-smokers, current smokers and former smokers were 74.3%, 13.2% and 12.5% in our study compared to 71%, 16% and 13% in 2011, respectively [18]. The prevalence of non-smokers and former/ex-smokers has increased while that of current smokers has decreased. The possible reasons for the decrease in the prevalence of current smokers were due to an increase in lung diseases among older smokers, and hence they decided to quit smoking besides the rigorous quit smoking campaigns launched by the Malaysian government. In addition, the prevalence of current smoker in our study was lower than the current smokers in South Korea, which was 17.4% [30]. This discrepancy was due to the operational definition in the South Korea study which had included both middle and older age group smokers in their study respondents.

The association between smoking status and social support observed in this study is noteworthy. In terms of smoking status, our study showed that former smokers had the highest prevalence of good social support, followed by non-smokers and current smokers. Lüscher et al., 2017 [31], and Scholz et al., 2016 [32], reported that social support interventions for smoking cessation were efficacious. Our former smokers have higher social support; hence, this would have contributed to their efforts in quiting smoking. There was a significant association between smoking and poor social support. This finding concurred with a local study published by Rusdi et al. in 2019 [21], which stated that current smokers have lower income and lower education level which probably contributed to the poor social support in them. Furthermore, cessation of smoking depends on good social support through behavioral interventions together with strong support from family and friends based on the literature [33]. Although not directly explored in this study, previous research has also reported that smoking has been linked to the heightened occurrence of social isolation and loneliness among older adults, indicating that smoking has negative effects on their psychosocial well-being [34].

## 5. Strength and Limitations

The strength of this study lies in its nationally representative health morbidity survey having a large sample size. Furthermore, this study utilizes well-established and validated measures to assess social support and smoking status. This strengthens its credibility and enhances the trustworthiness of the findings. Nevertheless, the cross-sectional study design does not allow to establish a causal relationship between social support and smoking status. Longitudinal studies would provide a more comprehensive understanding of the dynamic relationship between the predictors and social support among the elderly. Additionally, the data on smoking status were self-reported, which may cause a response bias. Hence, there may have been an underestimation or overestimation of smoking prevalence.

## 6. Conclusions

The findings of this study underscored the significance of age, education, income, and smoking status in shaping social support outcomes for this population. Specifically, this study revealed inadequate social support among older people who smoked currently, people with increased age, lack of formal education and low income. These findings could assist policymakers to develop targeted strategies and support systems at the national level to further enhance the social support network systems among the older smokers and to curb smoking, thereby promoting healthy aging in the Malaysian population.

## Figures and Tables

**Table 1 healthcare-11-02249-t001:** Sociodemographic characteristics of the participants (N = 3923).

Characteristics of Respondents	Total (n)	% (95% CI)
**Age**		
60–69 years	2523	66.5 (63.4, 69.2)
70–79 years	1095	26.0 (23.7, 28.4)
≥80 years	305	7.6 (6.4, 9.0)
**Gender**		
Male	1840	48.8 (47.1, 50.5)
Female	2083	51.2 (49.5, 52.9)
**Ethnicity**		
Malay	2580	58.1 (49.0, 66.6)
Chinese	705	26.6 (19.8, 34.7)
Indian	124	6.5 (4.0, 10.2)
Others	79	1.3 (0.6, 3.0)
Bumiputra Sabah and Sarawak	435	7.6 (4.3, 12.9)
**Marital status**		
Single/widow/widower/divorcee	1329	32.2 (29.6, 34.9)
Married	2591	67.8 (65.1, 70.4)
**Strata**		
Urban	1674	73.2(69.1, 77.0)
Rural	2249	26.8 (23.0, 30.9)
**Education Level**		
No formal education	788	14.4 (12.3, 16.8)
Primary education	1910	43.6 (39.3, 48.0)
Secondary education	961	32.4 (28.9, 36.0)
Tertiary education	264	9.7 (7.4, 12.5)
**Employment Status**		
Employed	1029	24.2 (22.1, 26.3)
Unemployed	2894	75.8 (73.7, 77.9)
**Income**		
Less than MYR 1000	2477	58.2 (54.4, 61.8)
MYR 1000–MYR 1999	839	21.5(19.1, 24.1)
MYR 2000–MYR 2999	318	11.2 (9.6, 13.0)
≥MYR 3000	245	9.2 (22.1, 26.3)
**Smoking Status**		
Non-smoker	2797	74.3 (72.0, 76.5)
Current smoker	605	13.2 (11.6, 14.9)
Former smoker	512	12.5 (11.0, 14.2)
**Living status**		
Living alone	291	6.3 (5.3, 7.5)
Not living alone	3632	93.7 (92.5, 94.7)

**Table 2 healthcare-11-02249-t002:** Association between social support with sociodemographic and smoking status.

Variables	Social Support				Crude OR, (95% CI)	*p*-Value	* AOR (95% CI)	*p*-Value
	Good		Poor					
	n	% (95% CI)	n	% (95% CI)				
**Age**								
60–69 years	1823	73.1 (69.3, 76.5)	689	26.9 (23.5, 30.7)	1.00		1.00	
70–79 years	688	65.0 (59.2, 70.4)	401	35.0 (29.6, 40.8)	1.46 (1.17, 1.82)	**<0.01**	1.18 (0.92, 1.50)	0.19
≥80 years	155	50.1 (41.7, 58.6)	149	49.9 (41.4, 58.3)	2.70 (1.95, 3.73)	**<0.01**	1.72 (1.19, 2.48)	**<0.01**
**Gender**								
Male	1297	72.7 (68.2, 76.8)	532	27.3 (23.2, 31.8)	1.0		1.00	
Female	1369	65.9 (61.6, 70.0)	707	34.1 (30.0, 38.4)	1.38 (1.12, 1.70)	**<0.01**	0.99 (0.77, 1.28)	0.94
**Ethnicity**								
Malay	1778	71.1 (66.9, 75.0)	791	28.9 (25.0, 33.1)	0.89 (0.58, 1.35)	0.56	1.15 (0.73, 1.81)	0.55
Chinese	450	65.8 (58.4, 72.5)	248	34.2 (27.5, 41.6)	1.13 (0.70, 1.83)	0.62	1.61 (0.95, 2.74)	0.08
Indian	87	64.4 (51.0, 75.9)	37	35.6 (24.1, 49.0)	1.20 (0.62, 2.34)	0.59	1.58 (0.79, 3.12)	0.20
Others	62	81.2 (66.4, 90.5)	17	18.8 (9.5, 33.6)	0.50 (0.21, 1.21)	0.12	0.59 (0.23, 1.55)	0.28
Bumiputra Sabah and Sarawak	289	68.5 (60.0, 75.9)	146	31.5 (24.1, 40.0)	1.00		1.00	
**Marital status**								
Single/widow/ widower/divorcee	797	60.0 (54.6, 65.1)	527	40.0 (34.9, 45.4)	1.86 (1.46, 2.38)	**<0.01**	1.44 (1.08, 1.91)	**0.01**
Married	1867	73.6 (69.4, 77.4)	711	26.4 (22.6, 30.6)	1.00		1.00	
**Strata**								
Urban	1161	69.9 (64.9, 74.4)	506	30.1 (25.6, 35.1)	1.00		1.00	
Rural	1505	67.5 (62.9, 71.8)	733	32.5 (28.2, 37.1)	1.12 (0.82, 1.51)	0.48	0.99 (0.77, 1.28)	0.92
**Education Level**								
No formal education	430	54.0 (48.0, 59.9)	357	46.0 (40.1, 52.0)	3.93 (2.33, 6.62)	**<0.01**	1.93 (1.13, 3.30)	**0.02**
Primary education	1301	68.4 (64.0, 72.6)	595	31.6 (27.4, 36.0)	2.13 (1.29, 3.53)	**<0.01**	1.19 (0.72, 1.95)	0.49
Secondary education	722	73.2 (67.1, 78.5)	236	26.8 (21.5, 32.9)	1.70 (1.09, 2.64)	**0.02**	1.24 (0.82, 1.87)	0.31
Tertiary education	213	82.2 (74.2, 88.1)	51	17.8 (11.9, 25.8)	1.00		1.00	
**Employment**								
Employed	751	75.7 (70.9, 80.0)	273	24.3 (20.0, 29.1)	1.00		1.00	
Non employed	1915	67.2 (63.1, 71.0)	966	32.8 (29.0, 36.9)	1.53 (1.23, 1.90)	**<0.01**	1.14 (0.89, 1.47)	0.29
**Income**								
Less than MYR 1000	1557	62.6 (58.1, 66.9)	911	37.4 (33.1, 41.9)	2.60 (1.52, 4.45)	**<0.01**	1.94 (1.21, 3.13)	**<0.01**
MYR 1000–MYR 1999	619	74.4 (69.4, 78.9)	215	25.6 (21.1, 30.6)	1.50 (0.88, 2.55)	0.14	1.28 (0.80, 2.06)	0.30
MYR 2000–MYR 2999	261	83.4 (77.6, 88.0)	56	16.6 (12.0, 22.4)	0.87 (0.49, 1.54)	0.62	1.24 (0.82, 1.87)	0.31
≥MYR 3000	203	81.3 (71.3, 88.4)	41	18.7 (11.6, 28.7)	1.00		1.00	
**Living status**								
Living alone	177	60.3 (51.8, 68.2)	113	39.7 (31.8, 48.2)	1.53 (1.03, 2.25)	**0.03**	0.96 (0.62, 1.48)	0.84
Not living alone	2489	69.8 (65.9, 73.5)	1126	30.2 (26.5, 34.1)	1.00		1.00	
**Smoking status**								
Non-smoker	1907	69.2 (65.3, 72.9)	883	30.8 (27.1, 34.7)	1.00		1.00	
Former smoker	353	73.6 (67.7, 78.7)	158	26.4 (21.3, 32.3)	0.67 (0.46, 0.97)	**0.04**	0.99 (0.74, 1.32)	0.95
Current smoker	406	65.1 (58.4, 71.2)	197	34.9 (28.8, 41.6)	0.83 (0.66, 1.04)	0.11	1.42 (1.05, 1.91)	**0.02**

* Multiple variable logistic regression analysis was performed after adjusting for other potential confounders in the model. No interaction was found among the independent factors (*p*-value > 0.05). Classification of table showed that the model correctly predicted 70.2% of the cases. Receiver operating characteristics curve analysis area under the curve was 0.64, *p* < 0.001.

## Data Availability

Data are available upon reasonable request. The dataset used and analyzed in this study is available from the National Institutes of Health, Ministry of Health Malaysia upon reasonable request and with permission from the Director General of Health, Malaysia.

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
