# Peer review of "Social Support among Older Persons and Its Association with Smoking: Findings from the National Health and Morbidity Survey 2018"

_healthcare, 2023, doi:10.3390/healthcare11162249_

Round 1

Reviewer 1 Report (Previous Reviewer 1)

I applaud the authors for addressing my comments.

Professional editing is recommended.

Author Response

Professional editing is recommended.

We thank you for the comments. Professional editing has been done as advised by the English language department of my university.

Reviewer 2 Report (New Reviewer)

The paper sounds good except for a few corrections/concerns:

1). Lines 71- 80 You need citations on several of the statements you have written to support your assertions.

2). Lines 83 - 84 You need a comparison group or reference group. For example, what were the figures for the adolescents in comparison to those of elderly people? This will help the reader to see the significance of what you have written in that sentence.

3). Lines 84 - 86 The statement is not clear, it is confusing and needs some clarification/redo it to make it more clearer. Seems to have some grammatical issues.

4). Line 114  Which women are you referring to when you say "these women"? The sentence needs restructuring.

5). Line 114 The word "Risk" should have all letters in small but the letter "r" here is written in caps.

6). Line 119 The word "under-reported" should be written as one with a hyphen as indicated here rather than two separate words.

Also need citation for the sentence on lines 119 - 120.

7). Lines 142 - 144 It is important to give a conceptual and/or operational definition of the term 'elderly'. This helps the reader to contextualize the specific target population for your study as these definitions vary from one country or context to the next.

8).Table 2 needs significant improvement on how the information is presented. It is so crowded. It is confusing to the reader.

Maybe give it a "far left align" to create space for the presentation of all the information in a more organized and spaced manner.

--------------------------------------------------------------------

Otherwise this is a very interesting topic and has far reaching implications in society as more people are living longer lives globally.

Author Response

Reviewer 2

Reviewer’s comments

Author’s response

1)

Line 71-80 You need citations on several of the statements you have written to support your assertions.Line 71-80

Thank you for highlighting this.

We have cited two references here to support my assertions in Line 74-77.

Social support encompasses emotional, instrumental, and informational assistance provided by social networks, including family, friends, and community [9]. It plays a vital role in promoting overall well-being, especially among older individuals [10].

References:

Phillips JE, Ajrouch KJ, Hillcoat-Nallétamby S. Key concepts in social gerontology 2010.

Tengku Mohd TAM, Yunus RM, Hairi F, Hairi NN, Choo WY. Social support and depression among community dwelling older adults in Asia: a systematic review. BMJ Open 2019; 9(7): e026667.

2).

Lines 83 - 84 You need a comparison group or reference group. For example, what were the figures for the adolescents in comparison to those of elderly people? This will help the reader to see the significance of what you have written in that sentence.

We agreed with your suggestion. We revised the sentences accordingly.

L84-86:

Social support and networking prevalence was found to be reduced among older persons, which was at 30.76% [9]. However, among Malaysian adolescents the prevalence of social support was higher at 55.4% [12].

Reference:
Yaacob, S. N., Mei Yee, S. F., & Wan, G. S. (2017). Predicting Role of Social Support and Academic Stress on Life Satisfaction among Malaysian Adolescents. Sains Humanika9(3-2). https://doi.org/10.11113/sh.v9n3-2.1269

3).

Lines 84 - 86 The statement is not clear, it is confusing and needs some clarification/redo it to make it more clearer. Seems to have some grammatical issues.

Thank you for the comment. I have made this sentence clearer as below.

L86-89:

Correspondingly, Mahmud et al, 2020 [13] found that having low income, being single, not having depression, absence of activities of daily living and having dependency in instrumental activities of daily living were important factors related to perceived social support among Malaysian older adults.

4)

Line 114  Which women are you referring to when you say "these women"? The sentence needs restructuring.

Line 115-117

Thank you for the comment. The sentence has been restructured.

Positive social support discourages smoking in middle-aged and older postmenopausal women these women [15].

5)

Line 114 The word "Risk" should have all letters in small but the letter "r" here is written in caps.

Line 120

Thank you for the comment. I have added the full stop and “Risk”  is in capital letter then.

…smokers. Risk

6).

Line 119 The word "under-reported" should be written as one with a hyphen as indicated here rather than two separate words.

Also need citation for the sentence on lines 119 - 120.

Line 125

In Malaysia, the prevalence of smoking in the older population was under-reported contrasting to smoking studies among adolescents [17].

Reference:

Lim KH, Teh CH, Heng PP, Pan S, Ling MY, Yusoff MFM, Ghazali SM, Kee CC, Shaharudin R, Lim HL. Source of cigarettes among youth smokers in Malaysia: Findings from the tobacco and e-cigarette survey among Malaysian school adolescents (TECMA). Tob Induc Dis. 2018 Nov 5;16:51. doi: 10.18332/tid/96297. PMID: 31516448; PMCID: PMC6659477.

7).

Lines 142 - 144 It is important to give a conceptual and/or operational definition of the term 'elderly'. This helps the reader to contextualize the specific target population for your study as these definitions vary from one country or context to the next.

Line 148-149

Thank you for the comment. We have added the statement below to contextualize the target population of my study.

Malaysia recognises persons aged 60 and above as elderly[9].

8).

Table 2 needs significant improvement on how the information is presented. It is so crowded. It is confusing to the reader

 Maybe give it a "far left align" to create space for the presentation of all the information in a more organized and spaced manner.

Otherwise this is a very interesting topic and has far reaching implications in society as more people are living longer lives globally.

Thank you for the comments.

We have reformatted the Table 2 to make it landscape in orientation and "far left align" to create space for the presentation of all information as advised.

Reviewer 3 Report (New Reviewer)

There were a number of key strengths with this paper. The method and results were well explained and presented and the overall conclusion was clear. There were some minor suggestions which I make below for these sections but otherwise the paper was generally well presented and written. The main area I think needs some further work is the rationale for the study. It wasn’t clear exactly how this paper was addressing a research gap.

In the introduction it was written that, “Several studies have reported a negative association between social support and smoking prevalence among older persons.”

Then in the next paragraph it was written that “Despite extensive research on the association between social support and health outcomes, there is a noticeable research gap when it comes to understanding the specific relationship between social support and smoking behavior among older adults.”

The references cited in the introduction seem to have already established the relationship between levels of social support and smoking among older adults. It wasn’t exactly clear what was missing in this research and what more needed to be addressed. Perhaps a few sentences explaining the limitations of the past research would have made it clearer exactly how this paper was making a unique contribution. Was it the role of some of the mediating factors that was the added value of this paper? The model tested in this paper had more factors that could have explained smoking? Or a different measure of social support that provided different insights? Please explain in greater depth what was missing from these previous research.

If this part of the paper can be strengthened it will also help the discussion.

In regards to minor changes and suggestions:

I agree with the authors that there isn’t the space to examine the conceptual and theoretical basis behind social support in this paper. However, they use the term loneliness later in the paragraph and this is a different concept to social support. So perhaps a few sentences outlining some of the key concepts might help and why you chose social support compared to other social well-being concepts to measure. And careful about using the terms loneliness and social isolation as these are potentially different concepts than social support.

Twice this sentence appeared in the abstract and introduction: “Overall, in Malaysia, the social support and networking prevalence was lower among 24 older persons at 30.76%.” There was no context explaining what the 30.76% figure meant, how it was measured etc.

This sentence needs qualification: “Social support is also associated with mortality, depression, and well-being in older people [4–8]” What direction are these relationships?

Lin 114 page 3 missing a full stop.

Method sentence needs fixing: “A respondent was identified as having poor social support if his/her scored less than 27 scores [25].”

I don’t think Table 1 is needed as Table 2 conveys much of that descriptive information anyway.

Unless you can find a reference for this sentence I would soften the language as it is speculative: “The reasons for the decrease in the prevalence of current smokers were probably due to an increase in lung diseases among older smokers, hence they decided to quit smoking besides the rigorous quit smoking campaigns conducted by the Malaysian government.” E.g. a possible reason… rather than saying it is probable that…

See previous comments.

Author Response

Reviewer 3

Reviewer’s comments

Response to reviewer

There were a number of key strengths with this paper. The method and results were well explained and presented and the overall conclusion was clear. There were some minor suggestions which I make below for these sections but otherwise the paper was generally well presented and written. The main area I think needs some further work is the rationale for the study. It wasn’t clear exactly how this paper was addressing a research gap.

In the introduction it was written that, “Several studies have reported a negative association between social support and smoking prevalence among older persons.”

Then in the next paragraph it was written that “Despite extensive research on the association between social support and health outcomes, there is a noticeable research gap when it comes to understanding the specific relationship between social support and smoking behavior among older adults.”

Thank you for the comments.

This comment was further explained in the text comparing research in other countries and Malaysia and the research gap in Malaysia was identified.

L98-100:
Despite extensive research on the association between social support and health outcomes in various countries, there is a noticeable research gap when it comes to understanding the specific relationship between social support and smoking behavior among older adults in Malaysia.

The references cited in the introduction seem to have already established the relationship between levels of social support and smoking among older adults. It wasn’t exactly clear what was missing in this research and what more needed to be addressed. Perhaps a few sentences explaining the limitations of the past research would have made it clearer exactly how this paper was making a unique contribution. Was it the role of some of the mediating factors that was the added value of this paper? The model tested in this paper had more factors that could have explained smoking? Or a different measure of social support that provided different insights? Please explain in greater depth what was missing from these previous research.

If this part of the paper can be strengthened it will also help the discussion.

Thank you for highlighting this.

There was no past research in Malaysia on this topic, hence this paper will be useful.

L97-99:
Despite extensive research on the association between social support and health outcomes in various countries, there is a noticeable research gap when it comes to understanding the specific relationship between social support and smoking behavior among older adults in Malaysia.

In regards to minor changes and suggestions:

I agree with the authors that there isn’t the space to examine the conceptual and theoretical basis behind social support in this paper. However, they use the term loneliness later in the paragraph and this is a different concept to social support. So perhaps a few sentences outlining some of the key concepts might help and why you chose social support compared to other social well-being concepts to measure. And careful about using the terms loneliness and social isolation as these are potentially different concepts than social support.

This paper only looked at social support and hence this is our limitations. The term loneliness and social isolation are removed.

Twice this sentence appeared in the abstract and introduction: “Overall, in Malaysia, the social support and networking prevalence was lower among 24 older persons at 30.76%.” There was no context explaining what the 30.76% figure meant, how it was measured etc.

We edited in the abstract as follows:
Overall, in Malaysia, the social support and networking prevalence was low among older persons at 30.76%.

In the introduction:
L84-85:
Social support and networking prevalence was found to be reduced among older persons, which was at 30.76%.

It was measured using the Duke Social Support Index (DSSI).

This sentence needs qualification: “Social support is also associated with mortality, depression, and well-being in older people [4–8]” What direction are these relationships?

L69-L71
Good social support is also associated with reduced mortality, reduced depression, and improved well-being in older people.

Lin 114 page 3 missing a full stop.

L120: We added the full stop

Method sentence needs fixing: “A respondent was identified as having poor social support if his/her scored less than 27 scores [25].”

We agreed and have corrected as below:
L198-199:
A respondent was identified as having poor social support if his/her score was less than 27.

I don’t think Table 1 is needed as Table 2 conveys much of that descriptive information anyway.

Table 1 highlights social demographic of the participants whereas Table 2  highlights association of social support with social demographic and smoking status. Hence both tables are required for this manuscript.

Unless you can find a reference for this sentence I would soften the language as it is speculative: “The reasons for the decrease in the prevalence of current smokers were probably due to an increase in lung diseases among older smokers, hence they decided to quit smoking besides the rigorous quit smoking campaigns conducted by the Malaysian government.” E.g. a possible reason… rather than saying it is probable that…

Thank you for your suggestion. We follow accordingly.

L325-328:
The possible reasons for the decrease in the prevalence of current smokers were due to an increase in lung diseases among older smokers, hence they decided to quit smoking besides the rigorous quit smoking campaigns conducted by the Malaysian government.

Round 2

Reviewer 3 Report (New Reviewer)

The authors have addressed the comments well and the paper seems ready for publication

The language quality again seems fine but probably requires proof reading and an editing check.

This manuscript is a resubmission of an earlier submission. The following is a list of the peer review reports and author responses from that submission.

Round 1

Reviewer 1 Report

The paper address an important problem.

There are a number of grammar errors and awkward sentences.

p1, ln. 29: "were collected were"; ln.45-46: "advancing age".

p.2, ln.82: "With relevance to"; ln.96: "missing data(.) The"; p.3, ln.143: "Malays following"; p.8, ln.199: "reduced", ln.200: "an increase in diseases in older respondents hence they quit smoking and the rigorous quit smoking campaigns". 

p.3, ln.136: more details on the statistical analysis should be given. How was "adjusting for other confounders' performed?

A reference for Hosmer Lemeshow test (p.3, ln.136) should be given.

p.8, ln.198: the three numbers do not add up to 100%. 

There are a number of grammar errors and awkward sentences. Professional editing is encouraged.

p1, ln. 29: "were collected were"; ln.45-46: "advancing age".

p.2, ln.82: "With relevance to"; ln.96: "missing data(.) The"; p.3, ln.143: "Malays following"; p.8, ln.199: "reduced", ln.200: "an increase in diseases in older respondents hence they quit smoking and the rigorous quit smoking campaigns". 

Reviewer 2 Report

The study explored the association between smoking and social support among a sample of older Malaysians. My comments are listed below:

1. The focus of the study was on social support and, yet, the authors did not explain what social support is. The concepts and theories behind social support are missing. No elaborations on why social support matters among older adults. 

2. No discussions on the measurement of social support. The study used the Duke social support index. The reader should be informed of what does the index really capture? Are there other measures of social support? The authors did not discuss the validity and reliability of the instrument either.

3. No elaborations on the findings. For example, why did former smokers have higher social support? Readers expect to understand the study findings.